# The Roles of Luteinizing Hormone, Follicle-Stimulating Hormone and Testosterone in Spermatogenesis and Folliculogenesis Revisited

**DOI:** 10.3390/ijms222312735

**Published:** 2021-11-25

**Authors:** Olayiwola O. Oduwole, Ilpo T. Huhtaniemi, Micheline Misrahi

**Affiliations:** 1Institute of Reproductive and Developmental Biology, Department of Metabolism, Digestion and Reproduction, Imperial College London, Hammersmith Hospital Campus, London W12 0NN, UK; o.oduwole@imperial.ac.uk; 2Institute of Biomedicine, University of Turku, 20520 Turku, Finland; 3Unité de Génétique Moléculaire des Maladies Métaboliques et de la Reproduction, APHP Hôpitaux Universitaires Paris-Saclay, Hôpital Bicêtre, Faculté de Médecine Paris Saclay, 94275 Le Kremlin-Bicêtre, France; 4Hôpital Paul Brousse, Université Paris Saclay, UMR-S 1193, INSERM, 94800 Villejuif, France

**Keywords:** spermatogenesis, folliculogenesis, FSH, LH, testosterone, mutation, intratesticular testosterone, azoospermia, Sertoli cell, Leydig cell, minipuberty, knock-out mice

## Abstract

Spermatogenesis and folliculogenesis involve cell–cell interactions and gene expression orchestrated by luteinizing hormone (LH) and follicle-stimulating hormone (FSH). FSH regulates the proliferation and maturation of germ cells independently and in combination with LH. In humans, the requirement for high intratesticular testosterone (T) concentration in spermatogenesis remains both a dogma and an enigma, as it greatly exceeds the requirement for androgen receptor (AR) activation. Several data have challenged this dogma. Here we report our findings on a man with mutant LH beta subunit (LHβ) that markedly reduced T production to 1–2% of normal., but despite this minimal LH stimulation, T production by scarce mature Leydig cells was sufficient to initiate and maintain complete spermatogenesis. Also, in the *LH receptor* (*LHR*) knockout (LuRKO) mice, low-dose T supplementation was able to maintain spermatogenesis. In addition, in antiandrogen-treated LuRKO mice, devoid of T action, the transgenic expression of a constitutively activating *follicle stimulating hormone receptor* (*FSHR*) mutant was able to rescue spermatogenesis and fertility. Based on rodent models, it is believed that gonadotropin-dependent follicular growth begins at the antral stage, but models of FSHR inactivation in women contradict this claim. The complete loss of FSHR function results in the complete early blockage of folliculogenesis at the primary stage, with a high density of follicles of the prepubertal type. These results should prompt the reassessment of the role of gonadotropins in spermatogenesis, folliculogenesis and therapeutic applications in human hypogonadism and infertility.

## 1. Introduction

Almost 15% of couples are infertile [1] and a male factor is involved in nearly half of the cases, making male infertility and subfertility a public health problem [2]. In addition, a decrease in the quantity and quality of sperm is now observed globally [3]. In order to institute adequate therapies, it is essential to broaden our knowledge of the mechanisms underlying the initiation and maintenance of the poorly understood process of spermatogenesis and the pathogenesis of its disturbances.

The regulation of spermatogenesis encompasses a complex array of endocrine, paracrine and metabolic interactions that involve Sertoli, Leydig, peritubular and germ cells, maintaining the proliferation and differentiation of spermatogenic cells [4,5,6,7]. The most important of these regulatory pathways involves the hypothalamic-pituitary-gonadal (HPG) axis, which coordinates essential functions through the action of the heterodimeric glycoproteins, follicle-stimulating hormone (FSH) and luteinizing hormone (LH) and the maintenance of high intratesticular testosterone (ITT) concentration. These two hormones are composed of a common α subunit and a hormone-specific β subunit. LH controls the production of testosterone (T) by Leydig cells, the endocrine cells located in the interstitium of the testis (Figure 1A). T is essential for male virilization and, in combination with FSH, it triggers and maintains spermatogenesis. The joint action of T and FSH is exerted on Sertoli cells, which delimit the wall of the seminiferous tubules that support germ cells undergoing progressive development into mature sperm (Figure 1A). According to the well-established concept, the initiation and maintenance of spermatogenesis in humans require increased secretion of LH during puberty, resulting in the multiplication and maturation of a significant number of Leydig cells, a prerequisite for the maintenance of high ITT. In fact, ITT is ~100 times higher than peripheral blood levels in normal men, and it is considered necessary for both virilization of the male external genitalia the maintenance of spermatogenesis.

The role of gonadotropins in the ovary is also controversial. Up to now, the role of FSH in the ovary is thought to start very late, from the antral stage onwards, because most available texts refer to rodent instead of human models. It is thus important to highlight the differences between human and animal models, and to be cautious with comparisons. The clarification of the role of FSH in humans, based on the observations of complete FSHR inactivation, is thus important. By contrast, our findings support the concept that in humans, FSH seems mandatory for the initiation of follicular growth beyond the primary stage. These findings contrast with the traditional view of an initial gonadotropin-independent follicular growth prior to the preantral and early antral stages. The persistence of a high ovarian reserve in these patients may be due to the lack of recruitment and growth of small follicles after the primary stage during reproductive life, as observed in prepubertal ovaries, due to the complete block of FSHR function.

While the principles of the hormonal regulation of spermatogenesis and folliculogenesis are common knowledge, the recently acquired genetic information from human mutations, and from genetically modified mice, on the function of gonadotropin and gonadotropin receptor genes have advanced our understanding and challenged old dogma. The precise mechanisms relating to LH or FSH individually have now been demonstrated with gonadotropin subunit and receptor mutations in humans and in knockout (KO) mice. Therefore, to establish novel treatment strategies for impaired spermatogenesis and folliculogenesis, understanding the role of each gonadotropin in normal and pathological conditions is crucial.

In this article, we summarize the information on the gonadotropin-driven regulation of spermatogenesis and folliculogenesis, and present results from our studies on patients with genetic defects in LHβ and FSHR, as well as on genetically modified rodent models. Our findings elucidate new aspects of the regulatory mechanisms of spermatogenesis and folliculogenesis and challenge the hitherto recognized basic principles of their hormonal regulation and of the regulatory defects in testicular and ovarian disorders, as well as offering new approaches to the treatment of male and female hypogonadism.

## 2. Principles and Regulations of Spermatogenesis

Spermatogenesis is a sequential biological event supporting germ cell maturation in the testicular seminiferous tubules [8,9]. The process occurs in a stepwise fashion, and involves an interplay between several autocrine, paracrine, and other hormonal stimuli and nutrients that are supportive of the germ cells development, through the cellular mechanisms of mitotic development, the meiotic recombination of genetic materials, and morphological sperm maturation [10,11]. The development and maintenance of qualitative and quantitative spermatogenesis relies on the production of LH and FSH from the anterior pituitary gland in response to the gonadotropin-releasing hormone (GnRH) from the hypothalamus. GnRH stimulates gonadotropin synthesis and secretion by gonadotrope cells from the anterior pituitary in discrete pulses into the systemic circulation to control the development, maturation and function of the gonads [12]. LH stimulates the production of T, adjudged to be the cornerstone of effective spermatogenesis, secondary sexual characteristics and functions, as well as psychological and anabolic actions [4,5,13]. FSH, on the other hand, enhances T action by maintaining the supporting function of Sertoli cells on spermatogenesis. The crosstalk between these hormones is crucial for the dual functions of fertility and virility of the adult testis [5,14]. While T is considered the master switch of spermatogenesis, FSH is known to contribute to the quality and quantity of the sperm. The interplay between the different components and the subsequent direct negative feedback effects of T and inhibin B, produced by the Sertoli cells, are essential for the feedback regulation of GnRH and gonadotropins secretion in the maintenance of the homeostasis of the HPG axis [14,15,16].

At the testicular level, the direct action of FSH and LH are produced through specific G protein-coupled membrane receptors (GPCR), FSHR [17] and LHR (LHCGR in humans) [18,19,20], expressed respectively in the Sertoli cells of the testes and granulosa cells of the ovaries or in the Leydig cells of the testes and thecal and luteal cells of the ovaries [5,21,22,23,24]. Leydig cells are responsible for the production of T, which induces the functional responses necessary for spermatogenesis via the AR expressed in Sertoli cells [7]. Remarkably, *FSHR* and *LHR* share the same chromosomal localization on chromosome 2p21 [25,26], suggesting evolutionary duplication of a common ancestral gene. By contrast, the homologous *thyroid-stimulating hormone receptor* (*TSHR*) was dispersed in the genome during the course of evolution to 14q31 [25,26,27]. The *LHR* gene is composed of 11 exons [19] and the *FSHR* gene of 10 exons [17,28,29], the largest of which, exon 11 and 10, respectively, encode the entire transmembrane domain of the receptor. This is different from other genes encoding G protein-coupled receptors with short extracellular domains devoid of intron. The gonadotropin receptor genes might have been formed during evolution by a recombination between a gene encoding leucin rich repeats (present in the extracellular domain) and a protoreceptor gene devoid of introns encoding a GPCR [27,30,31].

## 3. Genetic Defects of the Gonadal Axis Have Advanced the Understanding of the Role of Gonadotropins in Human Reproduction

The predominant function of LH is the stimulation of T production through binding to the cognate receptor, LHR (LHCGR in humans) on the surface of the Leydig cells, with high ITT considered a prerequisite for sperm production and maturation. T exerts its action through the cognate AR, expressed by several testicular cell types (including the Sertoli and peritubular myoid cells), apart from the spermatogenic cell lineage [32,33]. Mutations in the *LHβ* and *LHCGR* genes, culminating in disorders of sexual development and reproductive function [34,35], highlight the critical role of this hormone and its receptor.

The study of the few patients with genetic abnormalities in gonadotropin action offers the opportunity to better understand human gonadal physiology. The impact of mutations in *LHR* and *FSHR*, and their cognate hormonal ligands, has helped identify certain syndromes and allowed better understanding of the physiological action of hormones on their target organs [36,37]. Complete inactivating mutations of LHβ were found in five male patients [38,39,40,41], all of whom were masculinized at birth, but lacked pubertal development in the absence of functional pituitary LH and subsequent T production. The spermograms showed an absence of spermatogenesis, while the testicular biopsy revealed the absence of Leydig cells or very rare residual immature Leydig cells and a blockage of germ cell maturation. These observations indicate that in humans, LH is not necessary for the differentiation of the male external genitalia. Rather, human chorionic gonadotropin (hCG) produced by the placenta, homologous to LH, allows the masculinization of the genitalia before birth by acting on LHCGR. However, after birth, LH is essential for testicular maturation, in particular for the proliferation and maturation of Leydig cells, which secrete T for pubertal virilization and the initiation of spermatogenesis. However, in an unexpected observation in men with a genetic defect in LH even after several years of treatment with high doses of hCG, spermatogenesis was rare or absent in these patients who remain infertile, indicating a situation whereby an essential timely “window of testicular susceptibility” had been missed [42], which cannot subsequently be replaced. Hence, the capacity for proliferation and differentiation of Leydig cells is permanently impaired. Fertility, or normal spermatogenesis, has been restored by hCG treatment in two patients with *LHβ* mutation [43,44].

## 4. Proliferation of Testosterone-Producing Leydig Cells in Different Waves

The three waves of Leydig cell proliferation in humans are indicated in Figure 1B [7,13]. Each of these waves is characterized by the production of T. The first wave is antenatal, occurring after 10 weeks of fetal life and is hCG-dependent. Fetal T secretion peaks at 14–15 weeks [45], and induces the differentiation of the internal genitalia and masculinization of the external genitalia. The other two waves of Leydig cell proliferation are postnatal, regulated by pituitary LH. At the end of the second month of life, in the postnatal period, the HPG axis is transiently activated. The increased secretion of LH and FSH causes the development of a second wave of Leydig cells, and the appearance of a peak of T at levels close in magnitude to those observed at puberty. This event, called “mini-puberty”, is short-lived, and its role is poorly understood. At the end of the first year, the testes enter into a quiescent state. The third wave occurs at puberty, when the HPG axis is reactivated. Under the action of gonadotropins, the secretion of T is activated, virilization occurs, and spermatogenesis is triggered and maintained by the joint action of T and FSH.

To date, treatment of most adult patients with inactivating *LHβ* mutation has been ineffective, highlighting the lack of understanding of the mechanisms that trigger and maintain spermatogenesis in humans. In fact, the hormonal thresholds necessary for fertility in men are not known.

## 5. A New Concept of the Role of LH and T in Spermatogenesis

While the study of patients with complete gene inactivation allows for the understanding of the physiological role of the corresponding hormones, the study of partial gene inactivation offers a unique opportunity to study hormonal thresholds necessary for reproductive functions.

The study of a patient with a partial genetic defect in *LHβ* yielded unexpected information [46]. The 43 year-old proband (Subject IV-3, with a 46, XY karyotype) (Figure 2A and Table 1) was born in Morocco to consanguineous parents.

The patient consulted for absence of pubertal development and hypogonadism [46]. He had undetectable LH, very low circulating T levels, an absence of puberty and, consequently, an absence of spontaneous virilization in adulthood. FSH blood concentration was high. This evoked a picture of isolated LH deficiency. Surprisingly, however, several spermograms in this patient showed quantitatively and qualitatively normal-to-subnormal spontaneous spermatogenesis. In this individual, the molecular study by direct genomic Sanger sequencing found a homozygous deletion of nine bases in *LHβ*, yielding a deletion of amino acids 10 to 12 (His, Pro, Ile) localized in the amino-terminal part of the hormone, which allowed conformational changes (Figure 2B).

The in vitro study of the mutated LH, through the co-transfection of an expression vector encoding the mutated LHβ subunit and the α subunit in eukaryotic cells, showed that the expression and secretion of the mutated hormone were reduced. The heterodimerization of this mutated β subunit with the α subunit was detected by co-immunoprecipitation studies of the two subunits, but in a much-reduced amount compared to the heterodimerization of wild-type subunits. The mutated dimeric hormone retained residual activity and binds to the LHR, as shown by the activation of cellular adenylate cyclase, but its biological activity in vitro is greatly reduced (~1%) compared to that of the wild-type hormone (Figure 3).

A testicular biopsy showed heterogeneous seminiferous tubules separated by fibrous tissue (Figure 4A), containing few mature, vacuolated Leydig cells (Figure 4B,C). Half the tubuli were hyalinized or hypoplastic, with a thickened lamina, immature Sertoli cells, and spermatogonia. The other half comprised mature Sertoli cells and germ cells at all stages of differentiation (Figure 4A–C).

The in vivo activity of the mutated LH was demonstrated by studying the testicular biopsy of the patient. Immunocytochemical studies using antibodies specific for enzymes of the T biosynthetic pathway allowed the detection of very rare mature Leydig cells, comprising the enzymatic equipment necessary for T production (Figure 4). Indeed, there was a small number of typical mature Leydig cells expressing enzymes necessary for androgen synthesis, including cytochrome P-450 (CYP) 17α-hydroxylase (Figure 4D), 3β-hydroxysteroid dehydrogenase (3β-HSD) (Figure 4F), and CYP cholesterol-side-chain cleavage enzyme. This demonstrates the in vivo action of LH, the only hormone capable of inducing the expression of these enzymes, to stimulate T synthesis in the testis. We also detected fibroblast-like fetal Leydig precursors, located outside the tubular basal lamina, expressing 3β-HSD (Figure 4G) but devoid of CYP 17α-hydroxylase. There were markedly fewer Leydig cells in the patient’s testis tissue sample than in the corresponding age-matched control (Figure 4E).

Interestingly, for the first time, the demonstration of the presence of the LHCGR in these precursor Leydig cells was performed on a frozen testicular sample. This confirms that postnatal Leydig cells precursors are under LH control. This patient’s testicular histology thus differed from that of previously described patients with LHβ inactivation who had germ cell maturation defects and no mature functional Leydig cells [37].

The patient’s Sertoli cells expressed anti-Müllerian hormone (Figure 4H), the persistence of which indicates low levels of intratesticular T action, whereas the control cells demonstrated an absence of anti-Müllerian hormone (Figure 4I). The expression patterns of the AR (Figure 4J,K) and of both markers of germ-cell-maturation histone H1 (Figure 4L) and proacrosin (Figure 4N) were similar in the testicular samples from the patient and the control (Figure 4M,O).

Finally, the ITT concentration was measured in a sample of testicular biopsy of the patient using gas chromatography-mass spectrometry. In the patient, it was 20.2 ng/g of tissue, which still is 2–3-fold higher than normal peripheral T levels, but only 5.5% of that in an age-matched control (356.6 ng/g of tissue). These results thus confirmed the immunohistochemical results on the presence of functioning Leydig cells in the patients’ testes.

The low intratesticular T in the patient resulted in very low levels of peripheral serum T (0.4 ng/mL), which explained the lack of virilization, as evidenced by the absence of spontaneous pubertal development. The small amount of T produced in the testis was, however, sufficient, in conjunction with high FSH, to produce a local action and induce the development of the seminiferous tubules and the complete maturation of the germ line (Figure 4).

There are several arguments to explain the partial in vivo LH action in this patient. Firstly is the presence of mature Leydig cells expressing the enzymatic equipment necessary for T synthesis. This finding demonstrates the LH action on Leydig cells, albeit at a reduced efficiency, as evidenced by their reduced number. Secondly is the presence of ITT levels clearly above the serum values in the patient’s testes, with a testis-to-serum gradient of 40, indicating local testicular T production. These values were lower than in controls, in keeping with the few steroidogenic mature Leydig cells. Lastly, the presence of normal spermatogenesis with complete germ cell maturation indicated local androgen production and action on Sertoli cells, which expressed AR (Figure 4J).

Thus, in this patient, the correlation between clinical, biological, histological immunocytochemical data and the molecular studies allowed the diagnosis of a partial selective defect of LH function. The ITT of this patient, while locally sufficient for spermatogenesis, was inadequate to maintain systemic T at sufficient levels to induce virilization, owing to the hormone dilution in the blood and extracellular spaces.

Minimal LH action, postnatally and at puberty, induced in this individual a small population of mature steroidogenic Leydig cells, capable of synthesizing detectable intratesticular T concentrations. This low LH-driven local production of T was sufficient, in synergy with FSH, for the onset of normal spermatogenesis, inducing full germ cell maturation through local action on adjacent seminiferous tubules. The T production was, however, insufficient to induce systemic virilizing effects.

Another conclusion, suggested by this case, and by the comparison with most other patients with persistent infertility and with a low sperm count even after hCG administration in adult age and full virilization, is the existence of an optimal window of LH exposure, even at low concentrations, the absence of which might alter overall testicular maturation, impairing the onset of normal spermatogenesis. This hypothesis is clinically relevant because it suggests the presence of a preferential window of hCG treatment in LH-deficient hypogonadic patients, after which it is more difficult to restore fertility. Possibly, an early hCG treatment mimicking the minipuberty lacking in hypogonadialinfants, may be beneficial for restoring full fertility in these patients.

In conclusion, this partial genetic defect of LHβ provides a unique opportunity to study the in vivo action of LH and T in humans. In contrast to previous dogma, we can conclude that in this patient, reduced levels of LH activity and low ITT were sufficient to trigger and maintain complete spermatogenesis in adulthood, even if they are incapable of inducing systemic virilization [46]. It may be that the ITT level was so high only due to the fact that the testis is the major site of its synthesis in males. High ITT is necessary to produce sufficient plasma T concentrations to allow the masculinization of the external genitalia after significant dilution in blood, but it is not necessary for spermatogenesis.

## 6. Testosterone and Spermatogenesis in Rodents: High Intratesticular Testosterone (ITT) Microenvironment Is Not Essential for Sperm Production

Further evidence of the critical role of LH/T in the initiation and maintenance of spermatogenesis has been obtained from animal models [47,48] and experimental approaches [49], in addition to observations of hypogondal human males [50,51]. In all, there is a consensus that there is an absolute requirement of T for spermatogenesis in most mammals, except for the photoperiod-dependent Djungarian hamster, whose spermatogenesis is FSH-dependent [52]. Several experimental approaches, including hypophysectomy, pharmacological Leydig cell ablation, inhibition of GnRH secretion or action by sex steroids and GnRH analogues, immunization against the hormone receptor or the contemporary approach utilizing knockout mouse models of LHR or LHB [47,48,53,54,55], and the GnRH-deficient hpg mouse model [56,57,58], have been utilized to abolish gonadotropin secretion and/or action, followed by selective replacement with LH/hCG or T. The targeted disruption of the *Lhr* gene for example, results in infertility, with the LH receptor KO (LuRKO) mice displaying highly elevated circulating LH, moderately elevated FSH, profoundly reduced circulating T and ITT, underdeveloped reproductive accessory glands, and arrested postnatal sexual development and infertility. The mice also have Leydig cell hypoplasia and the postmeiotic arrest of spermatogenesis [47,48]. Selective replacement with exogenous T, however, allowed spermatogenesis to proceed to completion [59,60].

In most species, the concentration of testicular T is one or two orders of magnitude higher than those of peripheral circulation [61,62], thereby making it possible to use exogenously administered T as a male contraceptive due to its suppressive effect on gonadotropins through enhanced negative feedback, followed by suppressed endogenous T production, ultimately leading to decreased ITT levels [13,14,63]. Studies on rodents showed that ITT concentrations below the normal high levels support complete spermatogenesis, with both qualitative and quantitative spermatogenesis reported at an ITT of 30% of that of control animals [64,65]. Transcriptome analysis of the testicular tissues from both LuRKO and T-treated LuRKO mice showed most of the defects in gene expression in the testes in the absence of LH action being largely corrected by T administration [66]. LuRKO mice treated with subcutaneous T implants demonstrated complete restoration of spermatogenesis [67,68]. Zhang et al. [59] also noted the initiation of spontaneous qualitatively complete spermatogenesis in 12 month-old LuRKO mice to be due to the residual LH-independent constitutive T production. Because the residual spermatogenesis was abolished by antiandrogen treatment, the findings support the contention that even very low concentrations of ITT are sufficient to maintain spermatogenesis. In one of our studies [68], we systematically highlighted that a high ITT is not a prerequisite for complete spermatogenesis, with normal qualitative and quantitative spermatogenesis achieved at a plasma and ITT concentration of approximately 5 nmol/L, which is effectively twenty times lower than the normal ITT concentration. This observation presents a potential drawback for numerous clinical male T contraceptive trials, especially with respect to uniform suppression of spermatogenesis. While ITT concentrations vary significantly among fertile men and reflect LH pulsatility [69], both the minimal and the “ideal” concentration of ITT for optimal spermatogenesis remain unknown. This is even more salient in the description of serum markers, such as insulin-like factor 3, which decreases in response to decreased gonadotropins levels [70]. The correlation of this parameter alongside T and ITT concentrations requires further investigation.

The LHβ KO mice were also completely infertile, with retarded growth of the male accessory reproductive organs, and spermatogenesis halted at the round spermatid stage as a consequence of defective steroidogenesis [53,55], even though their sexual differentiation and fetal gonadal development were normal, an indication that pituitary LH is not a prerequisite for fetal T production in rodents [71]. The phenotype of the LHβ KO male closely mimics that of humans, with normal early sexual differentiation. In humans, the T requirement for masculinization in utero is dependent on placental hCG stimulating LHCGR, while in mice there is no CG. This early masculinization is independent of LH/Lhr action in mice. Postnatally, LH requirement is critical with both humans and mice showing low T levels, arrest of spermatogenesis, Leydig cell hypoplasia, hypogonadism, and infertility.

The global and cell-specific deletion of AR in mice provides conclusive evidence for spermatogenic maturation arrest without the support of T [24,72,73]. Numerous mutations of the AR have been described (http://androgendb.mcgill.ca (accessed on 18 June 2019), with many characterized by disturbances in either male development and/or fertility [74,75], as with the AR knockout (ARKO) mice that are infertile [76,77], with spermatogenesis arrested at the pachetyne spermatid stage [78].

## 7. FSH Is an Important Stimulus for Sertoli Cells and Essential Regulator of Spermatid Maturation

FSH action is exerted solely on FSHR-expressing Sertoli cells, and the effect depends on their developmental status. FSH stimulates both prenatally and prepubertally the proliferation of Sertoli cells and thereby determines their ultimate number. In rodents and primates, this effect starts in the second half of gestation, following the onset of fetal pituitary FSH production [79,80] and the appearance of FSHR expression [79,81]. In the peripubertal period, the rising concentration of FSH triggers the second phase of Sertoli cell proliferation [81,82], while post-puberty, FSH essentially supports spermatid maturation and the maintenance of quantitative normal spermatogenesis [83,84]. In adulthood, circulating FSH concentrations determine the size of the seminiferous tubules and testes by the number of spermatogenic cells [85,86], and acts as a survival factor for premeiotic germ cells [87]. The absence of FSH and/or its cognate receptor leads to a reduction in Sertoli cell number and testes size compared to that in normal testicular development, but does not stop spermatogenesis [81,88,89]. The Sertoli cell number also determines sperm production capacity due to the fact that each Sertoli cell is able to support a given number of germ cells during the proliferative phase [81,88,90,91,92,93].

## 8. FSH Is Essential for Quantitative Spermatogenesis in Rodents

The FSH requirement for spermatogenesis is a consequence of its extensive effects on several Sertoli cell functions. Evidence for this is derived from studies of classical animal models, as well as FSH (FSHβ KO) and Fshr (Fshr-KO) null mice [94,95,96], that demonstrate normal development of the reproductive organs and are fertile, albeit with decreased testis size (Figure 5), which reflects the reduced Sertoli cell number and the capacity to nurture and support germ cell development [88,90]. The phenotype of the Fshr KO mice demonstrated more pronounced disturbance of genital development and spermatogenesis than the FSHβ KO mice, even though spermatogenesis was still preserved [95,96]. However, the GnRH-deficient hypogonadal (hpg) mice, devoid of both gonadotropins, are infertile as the germ cells do not progress beyond the early meiosis of round spermatids [56,97]. Consequently, treatment of these mice with FSH stimulates germ cell proliferation, and further induces spermatid formation [98]. Furthermore, FSH is known to stimulate mitotic and meiotic DNA synthesis, spermatogonia and preleptotene spermatocytes and to act as a survival factor through its action on Sertoli cells [87]. Furthermore, observations from chemically and hormonally treated rats showed the relevance of FSH in the early stages of spermatogenesis, shortly before the enhancement of the T effect [99,100,101]. In the absence of FSH or the cognate receptor, the germ-cell-to-Sertoli-cell ratio decreases [88,90], indicating that the reduction in the number of germ cells is not only a result of the decreased Sertoli cell numbers, but also due to the decreased ability to nurture germ cells. Together, these findings indicate that, in rodents, while FSH is essential for the pubertal development of the full complement of Sertoli cells, its absence results only in quantitatively decreased sperm output in the adult.

## 9. Relevance of LH and FSH in Human Spermatogenesis

The essential role of FSH in maintaining male fertility has been extensively reviewed [102], with both FSHR and FSHβ mutations described [103]. To date, only one inactivating FSHR mutation has been described in men, with those affected being homozygous brothers of homozygous Finnish women with inactivating FSHR mutation. Their masculine phenotype was inconspicuous. The men appeared to present with variably disrupted spermatogenesis ranging from normozoospermia to severe oligozoospermia, but none of them was azoospermic [83]. The mechanism of the receptor inactivation was found to be due to reduced targeting of the mutated receptor protein to the plasma membrane [83,104]. Of the five affected males, four were subfertile, with reduced testis size resembling the conditions in the FSH- and FSHR-deficient mice; nevertheless, two of the men had fathered two children each. The men had elevated circulating FSH levels. Residual FSH signaling by the mutated receptor cannot be completely ruled out. Indeed, a functional study of the mutated receptor in vitro demonstrated that a minute fraction of the mutant receptors were targeted to the cell membrane and were able to respond to stimulation with high concentrations of FSH, with cyclic AMP [104] and β-arrestin-mediated signaling [105]. There is also a possibility that chronically elevated FSH levels in affected men are able to produce the marginal activation of the small number of mutant FSH that reach the plasma membrane. This would be analogous to a recent finding of the response of a woman with inactivating LHCGR mutation to high-dose hCG treatment, despite in vitro evidence of a totally inactivated receptor [106]. Thus, conclusive information about the phenotype of men with complete inactivation of FSHR is still lacking.

By contrast, all five men discovered as isolated cases of inactivating FSHβ mutations were diagnosed with azoospermia and infertility [37,107]. Considering the fact that these men were of different ethnicities and backgrounds, the FSHβ mutations detected were independent of each other. Treatment with FSH in this condition may be beneficial, but no such attempts were able to drive spermatogenesis to completion in the men with FSHβ mutation [107]. Because FSH treatment at adult age is unable to restore full spermatogenesis, it is possible that FSH exerts some earlier organizing effects that cannot be compensated for after the sensitive developmental window.

## 10. The Role of FSH in the Human Ovary Revisited

In both the *FSHR* and *FSHβ* genes, a moderate amount of missense, frameshift and stop-gained mutations have been identified in more than 60,000 individuals on the ExAC database (http://exac.broadinstitute.org/gene/ENSG00000170820 (accessed on 15 November 2021); http://exac.broadinstitute.org/gene/ENSG00000131808 (accessed on 15 November 2021) [108]. These patients represent exceptional models from which to deduce the physiological actions of gonadotropins in vivo in humans.

Further evidence for the residual function of the mutated Finnish FSHR receptor is provided by the ovarian phenotype of women carrying this mutation. Although these patients did not respond to FSH stimulation [109], follicles up to the antral stage can be observed in their ovaries [30]. This is in contrast to the ovaries of patients with primary ovarian insufficiency and complete loss of FHSR function [36,110] suggesting some residual activity of this mutated FSHR which, however, is not able to maintain fertility. Indeed, in vivo models of patients with different mutations of the FSHR have been studied [110,111,112,113,114,115].

### 10.1. Human In Vivo Model of Complete FSHR Inactivation

The study of an vivo model of total FSHR inactivation in a female patient allowed us to revisit the role of the FSHR in the ovary. This French patient Paris 1 (Table 1 and Figure 6) had primary ovarian insufficiency, delayed puberty, primary amenorrhea, with very low plasma levels of estradiol and inhibin B. Stimulation with high doses of FSH had no effect. A novel homozygous mutation of the FSHR gene was found and functional studies showed that the mutated receptor was not properly targeted to the cell surface and unable to activate adenylate cyclase even at very high FSH concentrations [110]. Histological studies of the ovaries highlight a complete blockade of follicular growth after the primary stage (Figure 6). The follicular population consisted mainly of primordial and intermediary follicles (51% and 42% of the total follicular population), with few primary follicles (7%) observed. The mean number (±SD) of these follicles per square millimeter was 24.2 ± 4.6, thus markedly enhanced (~18 times) compared with the follicular density described in women after puberty.

### 10.2. Lessons Derived from Studying a Complete Molecular Alteration of the Human FSHR in Human Primary Ovarian Insufficiency

Firstly, colonization of the primitive gonad by ovogonies and formation of primordial follicles in the fetal ovary is possible in the human, even in absence of FSH activity, as in mice with FSHR inactivation [96].

Secondly, follicle development in the cycling human ovary has classically been subdivided into “basal” and “hormone-dependent”. The limits between both periods vary according to the study and the species under observation. FSH in women is thought to intervene in follicular maturation and selection, starting from the late preantral or early antral stage, whereas growth from the primordial to the late preantral stages is thought to be gonadotropin-independent [116,117,118].

These traditional notions have been strengthened by observations in genetically modified mice deficient in FSHR. Well-developed secondary follicles are also found in female *FSHβ* knockout mice (Table 1) [94]. Therefore, early follicular growth up to the stage of secondary follicles and the formation of a functional theca interna is independent of FSH in mice.

By contrast, our findings support the concept that in humans, FSH seems mandatory for the initiation of follicular growth beyond the primary stage. These findings contrast with the traditional view of an initial gonadotropin-independent follicular growth prior to the preantral–early antral stages. Thus, caution must be taken when conclusions are derived from a rodent model, in which the reproductive physiology is in part different from the human physiology.

The study of our patient demonstrates that total FSHR inactivation in humans causes infertility with an early block of follicular maturation remarkably associated with abundant small follicles, as in prepubertal ovaries. Th persistence of a high ovarian reserve in this patient may have been due to the lack of recruitment and growth of small follicles after the primary stage during the reproductive life, as observed in prepubertal ovaries, due to the complete block of the FSHR function.

Notably, the presence of numerous reserve follicles in the ovaries of these patients may open the way to in vitro follicular maturation and treatment of their infertility in the future [119].

### 10.3. Lessons Derived from Studying Partial Molecular Alteration of the Human FSHR In Vivo

Studying other glycoprotein hormone receptors, LHR and TSHR, has demonstrated the existence of partial mutations of GPCRs, associated with a less severe phenotypes [120,121]. We thus supposed that partial loss-of-function mutations of the FSHR also exist.

We detected two patients with FSHR mutations, resulting in a partial loss of function associated with a novel phenotype [111,112]. These patients had primary and secondary amenorrhea, respectively (Table 2), normal pubertal development and secondary sexual characteristics, as well as remarkably normal sized-ovaries, in which growing follicles were detected by ultrasonography that markedly differed from those observed in the patient with a complete phenotype [111,112]. Paris 2 demonstrated about a dozen follicles up to 3 mm in diameter and Paris 3 demonstrated several antral follicles up to 5 mm. The ovarian stimulation of Paris 3 with recombinant FSH caused a partial response with an elevation of circulating estradiol and an increase in follicular diameter from 5 to 8.5 mm. However, no changes in plasma estradiol and inhibin levels and in the aspect of the ovaries at ultrasonography were observed in Paris 2 after administering high doses of recombinant FSH. Thus, the clinical phenotype of Paris 2 was more severe than that of Paris 3.

Paris 2 demonstrated a Leu601Val substitution in the third extracellular loop of the receptor and an Asp224Val mutation in the extracellular domain (Figure 7). Each parent of the patient was heterozygous for one of the mutations. Paris 3 was a compound heterozygote with an Arg573Cys mutation in the third intracellular loop of the receptor and an Ile160Thr mutation in the extracellular domain (Figure 7). The father and unaffected sister were heterozygous for the Arg573Cys mutation, while the mother carried the Ile160Thr mutation.

Functional studies revealed that the total residual receptor function of both alleles of the FSHR in each patient was more altered in the case of Paris 2 than in Paris 3 (Figure 7 and Table 2).

Histological studies of the ovaries of both patients showed normal primordial, primary, and secondary follicles and a small number of small antral follicles, which reached a maximum diameter of 3 mm in Paris 2 and 5 mm in Paris 3 (Figure 7 and Table 2).

The residual FSHR function in these two patients could sustain follicular development up to early antral stages. Paris 3, with the least severe loss of FSHR function, demonstrated larger (5 mm) follicles at ultrasonography (Figure 7) than Paris 2 (3 mm).

These findings demonstrate a correlation between the clinical and ovarian phenotypes and the degree of alteration of receptor function in patients with loss-of-function mutations of the FSHR (Figure 7 and Table 2).

The comparison of clinical, biological, histological and molecular studies in our patients thus resulted in a greater understanding of the role of FSH in ovarian follicular maturation, particularly during early antral development in humans; few models are available to analyze this step of follicular development. It is thought by many that basal levels of gonadotrophins are required for the growth of follicles from the preantral to the selectable stage [116,118].

The studies of our patients have shown that only partial FSHR inactivation can sustain follicular development up to the early antral stage [111,112], with incremental levels of FSH or of receptor function being required for the normal growth of early antral follicles before selection. If FSHR stimulation cannot be achieved properly, the growth of antral follicles cannot continue until ovulation, the development of antral follicles stops and the maximal size reached is indicative of the degree of residual FSHR function. Ultrasonography highlighted several small antral follicles in the patients’ ovaries, all of which were arrested at a similar stage of development, depending on the severity of the FSHR molecular defect. By contrast, after selection, the growth of the dominant follicle can proceed to the ovulatory stage even in presence of decreasing FSH levels [118].

In conclusion, a continuum of alterations to FSHR function might exist, as in the case of the LHCGR and other GPCRs, with a continuum of phenotypes linked to the degree of impairment of receptor function, with even milder phenotypes than those described here, and more developed follicles. The presence of numerous antral follicles in the ovaries of these patients may allow treatment of their infertility by in vitro maturation [122], possibly using culture mediums with a high content of recombinant FSH associated with steroids and growth factors.

## 11. Influence of Excessive FSH Action on Testicular Function

In rodents, a strong positive correlation exists between circulating FSH and testis development, with decreased FSH or FSHR function leading to impaired spermatogenesis [85,86]. In such situations, the neonatal administration of FSH increased the Sertoli cell number and testes size in rats [123]. Men presenting with high FSH levels due to an underlying etiology of primary hypogonadism or a gonadotroph adenoma appear to possess normal testicular function [124,125], an indication that excessive FSH may not exert untoward effects in apparently healthy men.

FSHR mutations found in women with pregnancy-associated ovarian hyperstimulation syndrome consequent to the broadening specificity to hCG at high concentrations, did not have any effect on the reproductive health of their corresponding male siblings [36,126,127,128]. An identified activating/gain-of-function (GoF) FSHR-D567G mutation was found to maintain spermatogenesis, after hypophysectomy and without T therapy [129], suggesting that strong constitutive FSH stimulation may compensate for missing LH and reduced T action. Furthermore, a serendipitously discovered non-symptomatic male carrier of an FSHR-N431I mutation demonstrated complete spermatogenesis despite suppressed serum FSH [130]. In both situations, constitutive receptor activity was detected, explaining the dearth of ligand through feedback mechanisms. It was associated with markedly altered agonist-stimulated desensitization and internalization of the FSHR in the second patient. Although there is a paucity of information on excess FSH on male health, there is the possibility that high FSH levels have no pathophysiological response. This is in contrast to the activating LHR (LHCGR) gene mutations that lead to early-onset gonadotropin-independent precocious puberty in boys [131,132]. The transgenic mouse line expressing FSHR-D580H in Sertoli cells supports the benign nature of the GoF FSHR mutation in males [133]. Despite robustly induced cAMP production in the absence of a ligand [134], transgenic males present with normal testis development and function, and do not differ significantly in Sertoli or germ cell number or fertility from their WT littermates [133]. Curiously, females with activating LHCGR mutations appear to have no phenotype. This is supposed to be due to the necessary cooperation between the LHR and FSHR in females for complete steroidogenesis and to the desensitization mechanisms that occur in vivo. Recently, it was shown that knock-in mice expressing a constitutively activating LHR are infertile, presenting with anovulation, hormonal alterations and polycystic ovaries. Likewise, transgenic mice overexpressing the LH receptor in the female reproductive system spontaneously develop endometrial tumor [135,136].

## 12. LH/T Regulation of Spermatogenesis and Strong FSH Activation

In clinical practice, hypogonadotropic hypogonadism (HH) patients are valuable tools for the study of the roles of gonadotropins in spermatogenesis. Those presenting with secondary hypogonadism consequent to congenital HH, including Kallmann syndrome, can be treated with pulsatile GnRH administration or with FSH and LH [84,137,138]. Studies on mice have shown that with normal T exposure, spermatogenesis is possible in the absence of FSH [67,68], but the opposite has never been documented. The discovery of mutations in humans and the generation of animal models of FSH and FSHR defects have been used to enhance our knowledge of their effects on spermatogenesis.

To gain further insight into the role of FSH in spermatogenesis, we [133] took advantage of the mouse expression of a constitutively activating mutation of *Fshr* (*Fshr*-D580H) in Sertoli cells [134]. While the females developed various reproductive abnormalities, their male counterparts were apparently normal when compared to their WT littermates (Figure 8A,B), providing a possible explanation of why only two activating FSHR mutations [129,130] have been found in men.

The crossbreeding of Fshr-CAM male mice [134] with heterozygous LuRKO female mice [47] brought about the generation of a double-mutant Fshr-CAM/LuRKO mouse [133], with strong FSHR stimulation and marginal T production. Unexpectedly, the mutant Fshr-CAM reversed the azoospermia and infertility of the LuRKO mice to a near-normal phenotype (Figure 8C,D), with normal development of the testes and accessory reproductive organs, and completion of spermatogenesis, despite delayed puberty and small litter sizes. The absence of LHR in these mice notwithstanding, the serum and ITT concentrations recovered from the known basal values to ~20 and 40% of that in WT. This finding is in line with the findings in mice expressing Fshr-D567G CAM in the *hp*g background [139], in which the mutant FSHR brought about increased cAMP and T levels but was unable to rescue spermatogenesis. This indicates that a robust FSH action is a prerequisite for successful spermatogenesis in the absence of sufficient T production.

To confirm whether rudimentary Leydig cell T production was responsible for the observed spermatogenesis in the Fshr-CAM/LuRKO mice, we blocked T action through treatment with the potent antiandrogen, flutamide. As expected, T inactivation in the WT resulted in shrunken seminal vesicles and azoospermia (Figure 9A), similar to the observed phenotype in the Sertoli and peritubular myoid cell-specific AR knockout mice [24,72]. However, spermatogenesis continued unabated in the Fshr-CAM/LuRKO mice, despite identical flutamide treatment, without any deleterious effect on the testis size, apart from shrunken seminal vesicles (Figure 9B). This is a clear indication that the strong constitutive FSHR activation was able to compensate for the missing LH/T pathway and the androgen effect required for spermatogenesis.

In line with the flutamide treatment, the expression of the androgen-dependent Sertoli cell genes *Drd4, Rhox 5*, *Aqp8*, *Eppin* and *Gata 1* was decreased in the WT as a phenocopy of the LuRKO mice but was not suppressed in the Fshr-CAM/LuRKO mice (Figure 9C,D). This finding was unexpected, considering the fact that FSH and T have different mechanisms of action, with FSH acting through a GPCR, and T through the nuclear transcription factor, AR. Interestingly, however, it has been shown that overlapping signaling mechanisms exist between the two hormones [102,140], as both activate the MAP/ERK and CREB signaling cascades, which is shown to be crucial for spermatogenesis through a rapid T signaling mechanism [141], and an intensified Sertoli cell intracellular free Ca^2+^ [142,143]. This may thus explain the ability of the strong FSH action to substitute for the missing androgen effect. However, the incomplete quantitative restoration of spermatogenesis suggests the relevance of T in the qualitative and quantitative process of spermatogenesis.

## 13. Inference and Summary

The study of rodent models has been extremely useful in understanding the impact of gonadotropins on reproduction. Comparison between genetically modified mice with mutations hindering or exaggerating gonadotropin or receptor function in humans with comparable genetic abnormalities has demonstrated distinct differences that must be considered when transposing rodent data to humans.

The study of phenotypic effects in patients with genetic defects causing total or partial dysfunction of an isolated gonadotropin or of the corresponding receptor offers a unique opportunity to understand the isolated actions of these hormones in vivo. These observations allow the identification of the molecular pathology of reproductive syndromes of hitherto unexplained etiology and the respective physiological effects of LH and FSH on target organs. They make it possible to revisit the physiological actions of gonadotropins in spermatogenesis and folliculogenesis. Understanding the isolated effects of LH and FSH on gonadal development, hormone production, germ cell maturation, follicular development, ovulation, and implantation may offer new perspectives for the treatment of abnormalities in human sexual development and infertility, as well as for contraception in both sexes.

Contrary to previous dogma, we observed that in a patient with mutant LH [46], minimal LH activity leading to the minimal production of T by very rare functional Leydig cells, first during “mini-puberty” in the postnatal period and then during puberty, was sufficient to initiate and maintain complete spermatogenesis in adulthood. This observation has consequences for the management of certain types of male infertility. In fact, hormone replacement therapy in patients with congenital hypogonadism of pituitary origin is currently started at puberty. The very early management of these patients, during a “window of opportunity” [42] from the first year of life [144], and the institution of transient treatment with gonadotropins in order to reproduce a physiological “mini-puberty”, could possibly promote the initiation of spermatogenesis and the maintenance of fertility in adulthood. Knowledge of variations in the testicular hormonal microenvironment throughout life and their consequences on the initiation and maintenance of spermatogenesis in adults is essential. This will not only lead to the improvement of therapeutic strategies for male infertility that take physiology into account, but also to the development of efficient methods of male contraception, whose current failures are linked to the difficult control of a poorly understood process [46].

FSH function and regulation is an integral part of the HPG axis. To the extent that it has been studied in males, the effects appear to be similar in most mammalian species; essentially, they involve the regulation and maintenance of testicular spermatogenesis. Although FSH function and regulation are not obligatory requirements for the completion of spermatogenesis in rodents, their deficiency leads to a reduction in testis size and sperm quantity. In humans, distinct fertility phenotypes, ranging from azoospermia to oligozoospermia, have been described in carriers of the FSHB or FSHR mutations, respectively. The reason for the different phenotypes, especially in men carrying a defect in FSH production or action, remains unclear. The study of a larger number of patients is warranted because it seems that the inactivating FSHR function described in men may carry some minor residual activity [104].

The correlation between clinical, biological, histological and molecular studies is very important to infer the physiological action of a receptor/hormone in vivo [36,111]. It is very likely that other modifying genes or environmental factors may modulate the functional defect of receptors or hormones of the reproductive HPG axes. The resistance to the recovery of spermatogenesis in men with FSHβ mutations during FSH treatment [107] suggests the presence of an earlier susceptible period, a “window of opportunity”, in the action of FSH.

In the past decade, men with idiopathic hypogonadotropic hypogonadism have been treated with a combined therapy of FSH and hCG to compensate for the lack of the gonadotropins [145]. Consequently, as a result of this positive influence on spermatogenesis, many studies of variable outcome have been conducted on men with idiopathic spermatogenic failure [146,147,148]. FSH treatment has been shown to improve sperm motility and morphology, as well as DNA damage and fragmentation [149,150]. Our recent finding that strong FSH stimulation can replace T in the maintenance of spermatogenesis explains the mechanism behind the documented gonadotropin-independent spermatogenesis in the hypophysectomized male with activating *FSHR* mutation [129], with no detectable gonadotropins, but slightly higher levels of T than at post-castration. The findings in this man were phenocopied by our recently described Fshr-CAM/LuRKO mouse model, thus providing an explanation for the rare phenotype. Our findings may also explain the reason why the standard FSH treatment of 50–100 IU 2–3 times/week has not been effective in the treatment of idiopathic oligozoospermia [146]. Two recent clinical studies [151,152] have shown that a sufficiently long high-dose FSH treatment, of at least 150 IU every other day for a minimum duration of at least six months, significantly improves sperm parameters. Indeed, Ding et al. [152] showed improvement in spermatogenesis and pregnancy rates in a group of idiopathic oligozoospermic men, with the best results achieved using 300 IU FSH recombinant human FSH treatment every other day for 5 months. Furthermore, it is known that certain patient groups are more responsive to FSH treatment than others, depending on their genetic background [153] and other associated factors, suggesting the possibility of pharmacogenetic approaches to individualized infertility treatments.

In conclusion, we have shown in our study on a genetic defect of LH in a man with normal spermatogenesis and genetically modified mice that the dogma of high ITT requirement for spermatogenesis has to be revised. High ITT is not necessary for spermatogenesis, but it is necessary for the production of sufficient circulating T levels for masculinization of secondary sex characteristics as well as metabolic and central actions of sex steroids. A robust and constant stimulation of FSHR, on the other hand, may be beneficial to boost suppressed spermatogenesis.

In vivo models of FSHR loss-of-function mutations in humans allowed us to revisit the role of FSH in folliculogenesis. A comparison of clinical, biological, histological, and molecular studies is critical to differentiate partial and complete loss of FSHR function in vivo. Our studies support the concept that in women, FSH is mandatory for the initiation of follicular growth after the primary stage. The gonadotropin-dependent stage in humans starts as early as the primary stage. These results are different from what is observed in rodent species, in which the phase of gonadotropin-independent follicular growth is much longer until the preantral- and antral stages.

In patients with primary ovarian insufficiency and partial loss of FSHR function, folliculogenesis can proceed only up to the small antral stages. Incremental doses of FSH are needed for the growth of the antral follicle, and, if this cannot be achieved, there is a block in follicular growth, with the size of the antral follicles being correlated with the remnant mutated FSHR function.

These observations support the early use of FSH in vitro in humans to promote small follicular growth. Thus, care should be taken when conclusions are drawn from rodent models, whose reproductive physiology is partly different from human physiology.

In addition, the complete inactivation of the FSHR in women causes infertility with a paradoxical abundance of small follicles, whose density is close to that of the prepubertal ovaries, with a large number of reserve follicles. This may be caused by the absence of follicular recruitment during the reproductive life of these patients due to the inactive FSHR. The high small follicular density in the ovaries of such patients may lead to a future treatment for their infertility. Fertility preservation should be recommended by freezing ovarian fragments in order to avoid the rapid and irreversible loss of the follicular pool that occurs over time through follicular atresia and apoptosis. In vitro small follicular maturation or follicular activation could be carried out in future, as this would make possible the treatment of infertility in these patients [119].

## Figures and Tables

**Figure 1 ijms-22-12735-f001:**
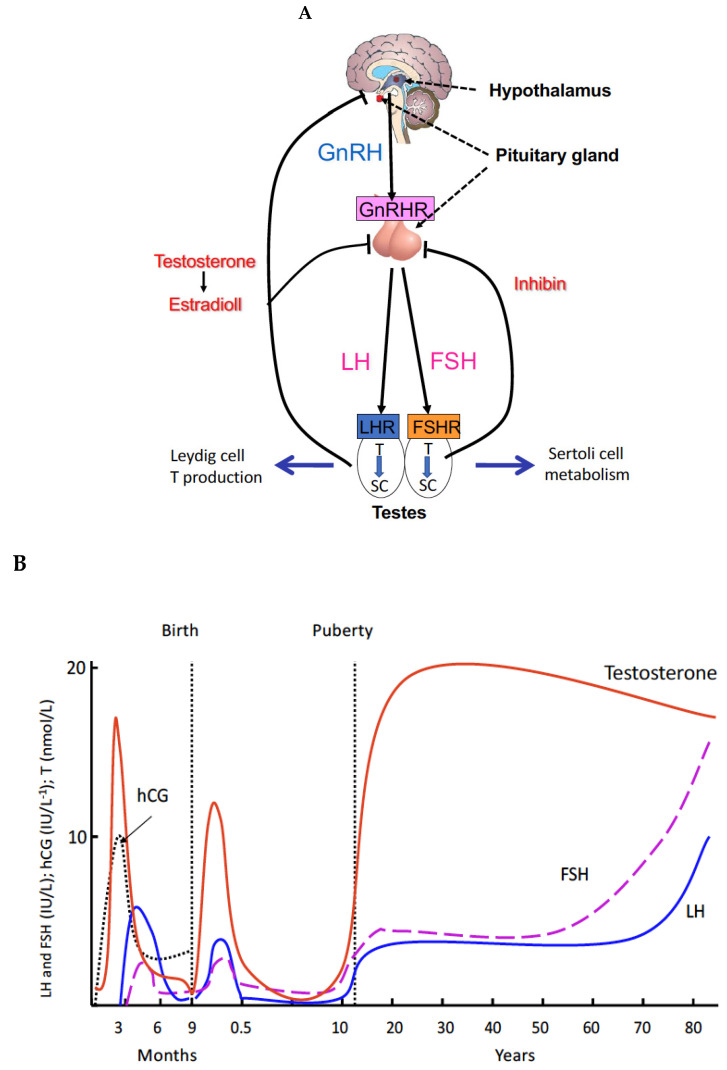
Control of spermatogenesis. (**A**) Hormonal control. Gonadotropin-releasing hormone (GnRH) produced by the hypothalamus stimulates, by activating its receptors (GnRHR), the synthesis and release of the gonadotropins LH and FSH in the anterior pituitary gland. LH induces the proliferation and maturation of interstitial Leydig cells which will secrete T. FSH acts on the Sertoli cells (SC) of the seminiferous tubules by stimulating the production of signaling molecules and metabolites necessary for spermatogenesis. In conjunction with T and FSH the Sertoli cells indirectly stimulate the proliferation and maturation of germ cells in the seminiferous tubules from which mature sperm are released to the seminiferous tubular fluid and transported to the epididymis for storage and final maturation. (**B**) The three waves of Leydig cell proliferation in humans. Each of them is characterized by the production of T. The first wave is antenatal, occurring after 10 weeks of fetal life and is hCG-dependent. Fetal T secretion peaks at 14–15 weeks, and induces the differentiation of the internal genitalia and masculinization of the external genitalia. The other two waves of Leydig cell proliferation are postnatal, regulated by pituitary LH. At the end of the second month of life, in the postnatal period, the HPG axis is transiently activated. The increased secretion of LH and FSH causes the development of a second wave of Leydig cells, and the appearance of a peak of T at levels close in magnitude to those observed at puberty. This event, called “mini-puberty”, is short-lived, and its role is poorly understood. The third wave starts at puberty and lasts for the rest of a man’s life.

**Figure 2 ijms-22-12735-f002:**
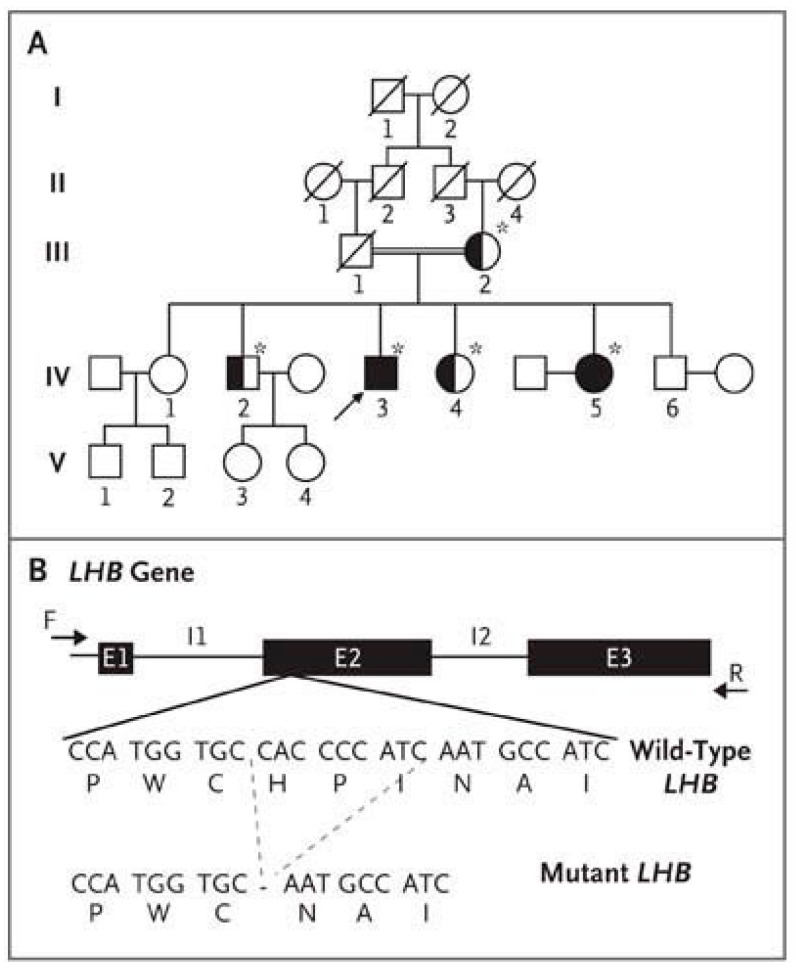
Pedigree of the patient with hypogonadism and *LHβ* Mutation. (**A**) The pedigree of the proband (Subject IV-3, arrow). The double line indicates consanguinity. For the family members who underwent genetic testing (denoted with an asterisk), solid symbols indicate homozygosity for the *LHβ* mutation, and half-solid symbols heterozygosity. (**B**) The complete coding region of the *LHβ* gene, including exons E1 through E3 and introns I1 and I2. The nine-base deletion in exon E2 results in the deletion of three amino acids from the protein. (For wild-type and mutant *LHβ*, the three-base codons are listed above the corresponding amino acid, represented by its single-letter symbol). From [46] with permission.

**Figure 3 ijms-22-12735-f003:**
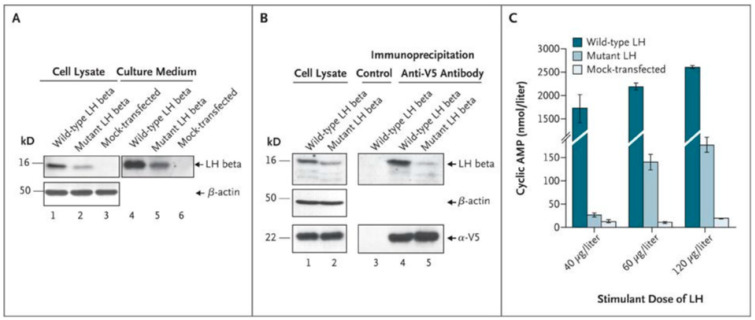
Production and heterodimerization in mutant and wild-type luteinizing hormone (LH) β and bioactivity of mutant and wild-type LH. Panel (**A**) *shows low intracellular and secreted levels of mutant LH beta subunit in transfected cells.* Western blot analysis was performed on cell lysate (lanes 1 through 3) and culture medium (lanes 4 to 6) of human embryonic kidney (HEK) 293T cells: cells producing the LHα subunit and the wild-type LH beta subunit (lanes 1 and 4), cells producing the LH alpha subunit and the mutant LHβ subunit (lanes 2 and 5), and mock-transfected cells (lanes 3 and 6). An anti–β-actin antibody was used as a loading control (lanes 1 through 3). Identical volumes of culture medium, concentrated by a factor of about 20 for wild-type LHβ (lane 4) and about 400 for mutant LHβ (lane 5), were loaded. Wild-type LHβ and mutant LHβ were immunodetected with an anti–hCGβ antibody (Abcam) displaying strong cross-reactivity. Panel (**B**) *shows low levels of dimerization of the alpha subunit and mutant beta subunit of LH.* Coimmunoprecipitation experiments were performed on cell lysates from COS-7 cells producing the α-V5 construct and either wild-type or mutant LHβ. Immunoprecipitation was performed with the use of anti-V5 antibody (lanes 4 and 5) or with a nonimmune immunoglobulin as a control (lane 3). This was followed by immunodetection of wild-type LHβ and mutant LHβ, with the use of a polyclonal anti–hCGβ antibody (Abcam), or of the α-V5 construct, with an anti-V5 antibody. An anti–β-actin antibody was used as a loading control (lanes 1 and 2). Panel (**C**) *shows markedly lower levels of mutant LH bioactivity in HEK 293 cells expressing the human LH receptor*. The secreted levels of wild-type LH were quantified by means of immunofluorometric assay and used to generate a dose–response curve. Comparative quantification of secretion of wild-type and mutant LHβ was carried out through Western blot analysis (Panel (**A**), reflecting one representative experiment). HEK 293 cells expressing the human LH receptor were stimulated with concentrated culture medium containing a similar amount of either wild-type LH or mutant LH beta. Concentrated culture medium from mock-transfected cells were used as a negative control. The mean results are shown for three independent experiments using each of three doses of mutant or wild-type LH to stimulate cyclic AMP production. I bar indicates standard deviations. From [46], with permission.

**Figure 4 ijms-22-12735-f004:**
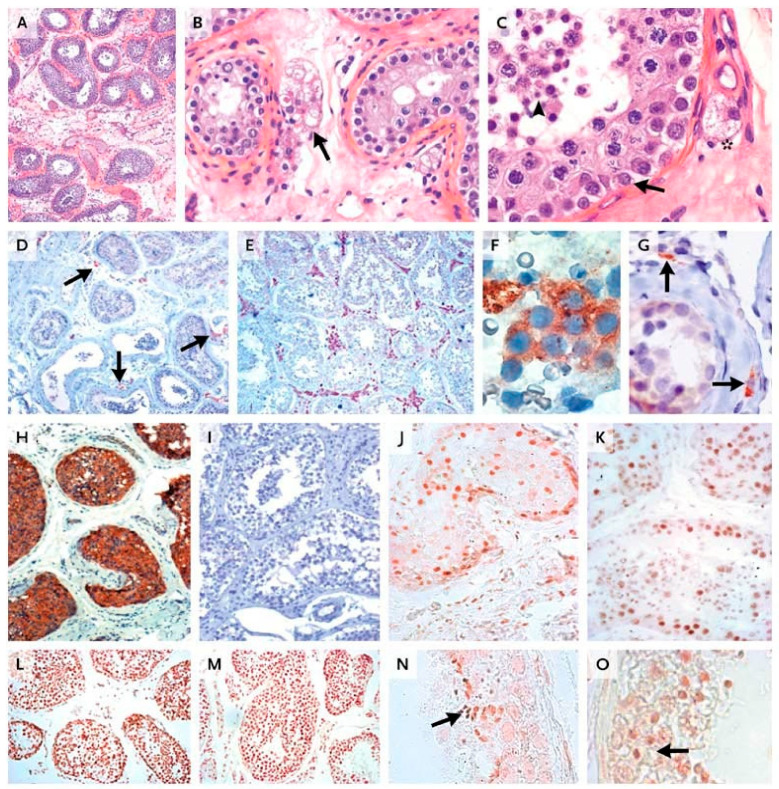
Histologic and immunocytochemical studies of the patient’s testis. (**A**) Seminiferous tubules separated by a fibrous interstitium. In Panel (**B**), a few vacuolated Leydig cells are visible (arrow). (**C**) shows all stages of germ-cell differentiation, from spermatogonia (arrow) to spermatozoids (arrowhead); there is also an isolated, mature Leydig cell in the interstitium (asterisk). In Panels (**A**–**C**), staining was performed with hematoxylin and eosin. In Panel (**D**), only a few interstitial androgen-producing cells positive for cytochrome P-450 (CYP) 17α-hydroxylase are visible (arrows), whereas such cells are abundant in a sample from an age-matched control (Panel (**E**)). We detected 3β-hydroxysteroid dehydrogenase expression in both the few interstitial mature Leydig cells also expressing CYP 17α-hydroxylase (Panel (**F**)) and in the fibroblast-like precursors, adjacent to the tubular basement membrane and lacking CYP 17α-hydroxylase (Panel (**G**), arrows). Panel (**H**) shows Sertoli cells strongly expressing anti-Müllerian hormone, whereas no such expression is observed in the control sample (Panel (**I**)). In Panels (**D**) through (**I**), staining was performed with hematoxylin. The expression pattern of the androgen receptor was identical in interstitial and intratubular cells from the patient (Panel (**J**)) and from the control (Panel (**K**)). A similar pattern of expression of histone H1, a marker of germ-cell maturation, was observed in maturing germ cells obtained from the patient (Panel (**L**)) and the control (Panel (**M**)). Proacrosin was detected in spermatids and spermatozoids from both the patient (Panel (**N**), arrow) and the control (Panel (**O**), arrow), indicating their advanced degree of maturation. Panels (**D**,**E**,**H**–**M**) are at the same magnification; Panel (**A**) is at a slightly lower magnification; Panels (**B**,**G**,**N**,**O**) are at twice the magnification; and Panels (**C**,**F**) are at four times the magnification. From [46], with permission.

**Figure 5 ijms-22-12735-f005:**
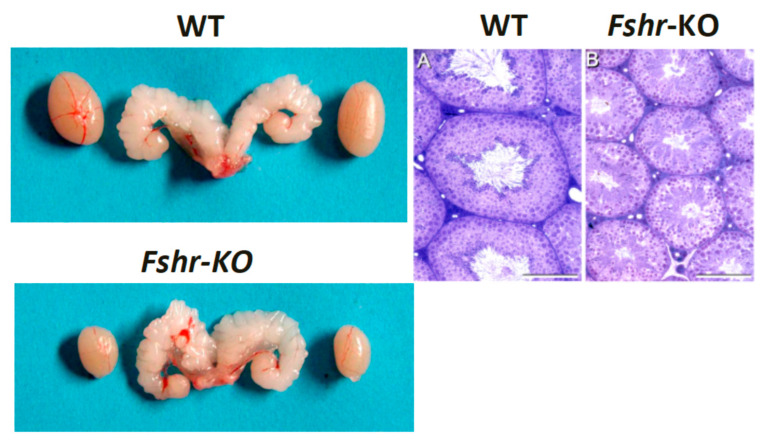
Testes and seminal vesicles of adult wild-type (WT) and FshrKO mice (**left panels**), and testicular histology of the same genotypes (**right panels**). No difference was observed in seminal vesicle sizes between the two genotypes, but the size of the FshrKO testes is about half that of the WT. Furthermore, while full spermatogenesis is visible in the histology of both testes, the tubular diameter is clearly narrower in the knockout testis. From Dr. Harry Charlton (University of Oxford), with permission.

**Figure 6 ijms-22-12735-f006:**
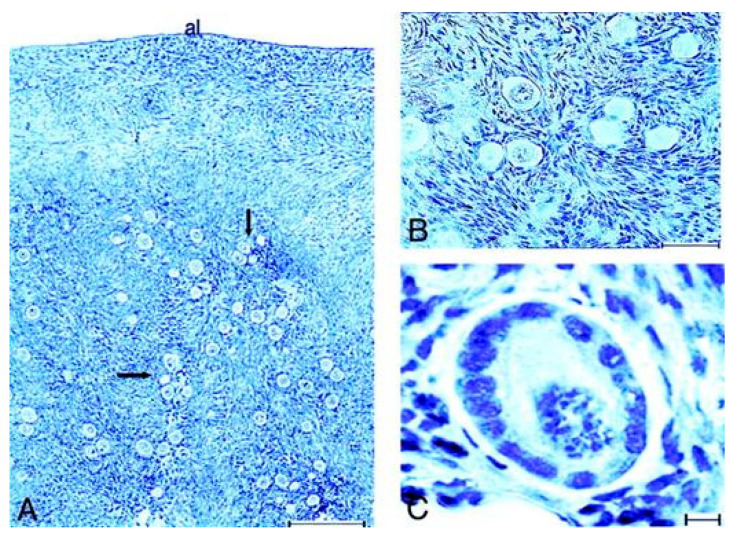
**Top:** Histology of the ovaries of the patient. (**A**,**B**), primordial and intermediary follicles in the deep cortical region (arrows). Magnification, ×40 and ×400, respectively. Albuginea. Bar, 200 (**A**) and 50 (**B**) μm. (**C**), primary follicle of normal histology, constituted of cubic granulosa cells surrounding an oocyte. Bar, 10 μm. From [110]. **Below:** Immunocytochemical study of the ovaries of the patient. Low magnification (×40) of the cortical region of the ovaries of the patient (**A**) and of a normal woman of comparable age (**B**), immunolabeled with an anti-c-Kit antibody. Note the difference in the follicular density and in the distribution of small follicles. Bars, 200 μm. (**C**), expression of c-Kit at the surface of the oocyte of an intermediary follicle in the patient’s ovary. Bar, 10 μm. (**D**), immunolabeling with anti-PCNA antibody. PCNA expression is detected in the granulosa cells and in oocytes of primordial and intermediary follicles in the patient’s ovary. The value, al, denotes Albuginea. Bar, 10 μm. (**E**–**G**), primordial, intermediary and primary follicles of the control ovary labeled with anti-PCNA antibody. The oocytes and some granulosa cells are immunolabeled. The follicles were not grouped in nests, but isolated in the cortical stroma. Bar, 10 μm. From [110], with permission.

**Figure 7 ijms-22-12735-f007:**
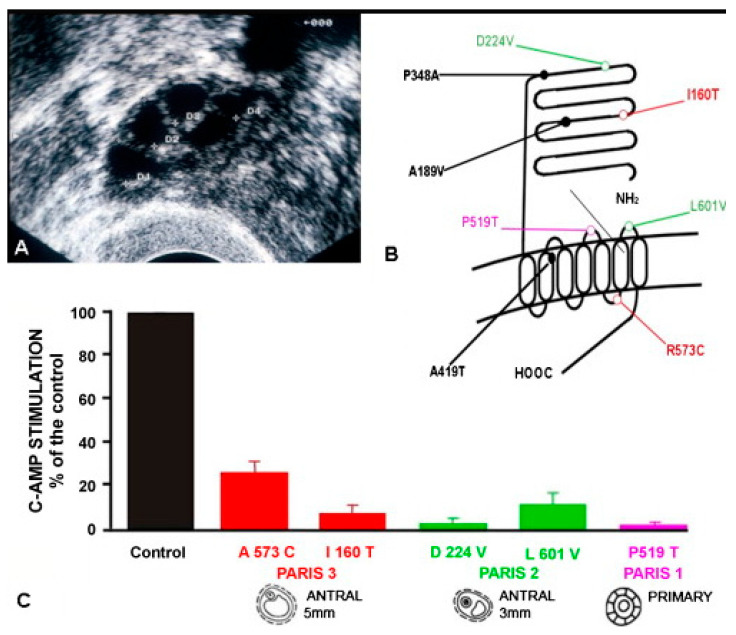
Loss-of-function mutations of the FSHR. (**A**) Ultrasonography of patient Paris 3 showing several antral follicles up to 5 mm diameter. (**B**) Localization of natural loss-of-function mutations of the FSH receptor. See text for references. (**C**) Comparative functional studies of loss-of-function mutations of the FSHR. The maximal stage of follicular maturation is illustrated below. From [36], with permission.

**Figure 8 ijms-22-12735-f008:**
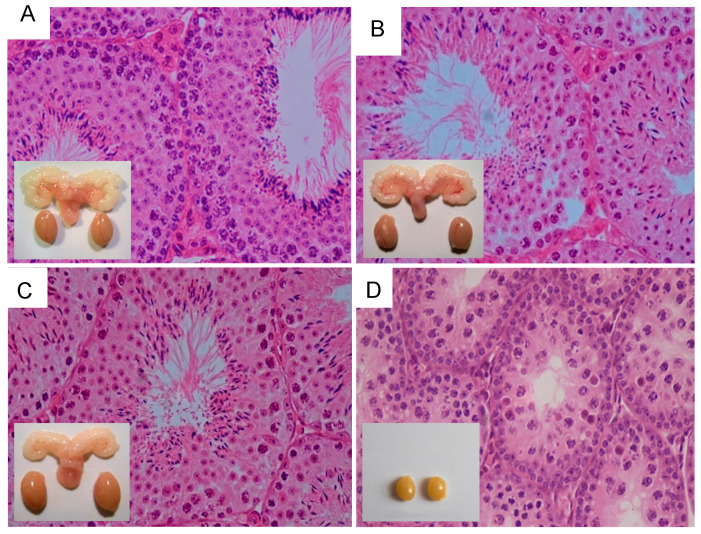
Testicular histology and macroscopic views of testes and urogenital blocks of different mouse genotypes: (**A**) WT, (**B**) Fshr-CAM, (**C**) Fshr-CAM/LuRKO, and (**D**) LuRKO mice. (**A**–**C**) show normal spermatogenesis and testis and seminal vesicle (SV) sizes. In (**D**), spermatogenesis is arrested at the round spermatid (RS) stage, with small testes and rudimentary seminal vesicle (not visible). Scale bars: 50 μm; 10 mm (insets). From [133], with permission.

**Figure 9 ijms-22-12735-f009:**
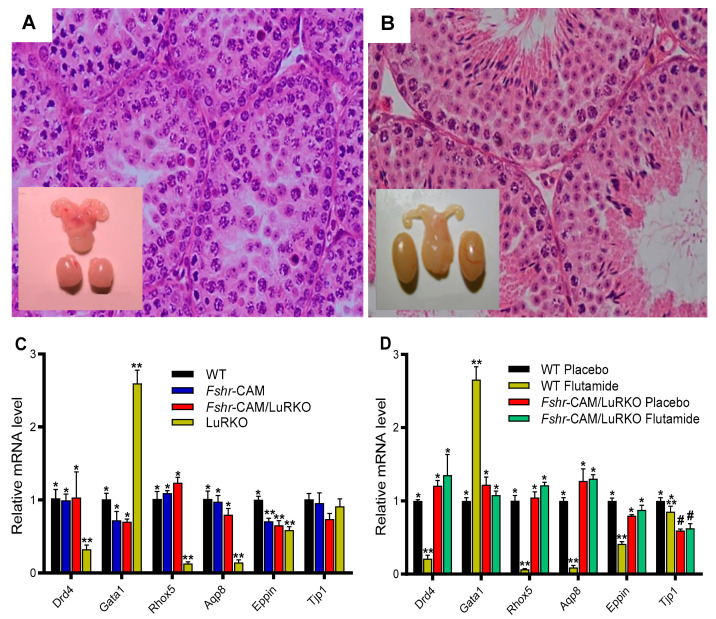
Effect of anti-androgen flutamide treatment on wild-type (WT) and genetically modified mice. (**A**,**B**): testicular histology and macroscopic views of the testes and urogenital blocks of WT and Fshr-CAM/LuRKO mice (**A**): the treatment arrested spermatogenesis at round spermatid stage in WT mice and reduced their testis and seminal vesicle sizes. (**B**) identical treatment of Fshr-CAM/LuRKO mice had no apparent effect on their spermatogenesis and testis size but reduced seminal vesicle size (arrows in (**B**)). (**C**,**D**): expression of selected target genes in untreated (**A**) and flutamide-treated (**B**) mice. (**A**): expression of androgen-regulated (*Drd5*, *Rhox5*, *Eppin,* and *Tjp1*), postmeiotic germ cell–specific (*Aqp8*), and germ cell–regulated (*Gata1*) genes in WT, Fshr-CAM, Fshr-CAM/LuRKO and LuRKO testes. (**B**): effect of flutamide treatment on expression of the same androgen-regulated genes in WT and Fshr-CAM/LuRKO mice. Data represent mean ± SEM. n = 3 samples/group. Bars with different symbols (*, **, #) differ significantly from each other (*p* < 0.05; ANOVA/Newman-Keuls). The remarkable finding is that while flutamide treatment suppressed the expression of strictly androgen-dependent genes in WT mice, the same effect was not observed in the testis of Fshr-CAM/LuRKO mice. Scale bars: 50 μm. From [133] with permission.

**Table 1 ijms-22-12735-t001:** Serum hormone levels in the proband with LHB mutation, his affected sister and members of their family.

Subject No. †	Sex	Age	LH		FSH		Testosterone	Androstene	17β-Estradiol	Estrone	Inhibin B	AMH	α-Subunit
			Baseline	Peak	Baseline	Peak							
		Yr	IU/L				ng/mL	μg/L	pg/mL	pg/mL	pg/mL	ng/mL	IU/L
IV-3 (proband)	M	43	ND	ND	20.7	29.8	0.48 *	0.62	<10	29	71.6	9.3	1.28
IV-5 (affected sister)	F	46	ND	ND	53.3				13				
IV-4 (unaffected)	F	32	4.5		4.6				66				
IV-2 (unaffected)	M	40	2.5		5.4		4.00						
III-2 (unaffected)	F	71	24.5		42.3				<10				
Normal range													
Male			0.8–7.6		0.7–11.1		3.30–10.00	0.2–2.90	<35	10–60	60.0–330.0	3.0–5.4	<0.70
Female			1.1–11.6	3.3–46.4	3.0–14.4	4.5–28.8			30–50 for follicular phase, 150–230 for luteal phase				

The peak value was the maximum level measured within 90 min of the intravenous administration of 100 μg of gonadotropin-releasing hormone. Serum levels of luteinizing hormone (LH) and follicle-stimulating hormone (FSH) were determined by means of an immunometric chemiluminescence assay (Siemens). The intraassay coefficient of variation was between 3.40% and 13.10% for LH (the latter value, at levels ≤0.15 IU per liter) and between 2.9% and 3.4% for FSH. The inter-assay coefficient of variation was between 6.2% and 23.9% for LH (the latter value, at levels ≤0.15 IU per liter) and between 4.1% and 4.8% for FSH. The sensitivity of the LH assay was 0.05 IU per liter. The T level was determined with the use of a radioimmunoassay (Orion Diagnostica). The intra- and inter-assay coefficients of variation were between 3.8% and 7.5% and between 4.8% and 7.0%, respectively. The sensitivity of the T assay was 0.029 ng per milliliter (0.100 nmol per liter) (limit of measurement, 0.140 to 14.400 [0.5 to 50.0]). To convert values for T to nanomoles per liter, multiply by 3.467. To convert values for androstenedione to nanomoles per liter, multiply by 3.492. To convert values for 17β-estradiol to picomoles per liter, multiply by 3.671. To convert values for estrone to picomoles per liter, multiply by 3.699. ND denotes not detectable. † Data for the proband, Subject IV-3, are from the initial laboratory assays performed during the first consultation, 3 months after the last T injection. Subject IV-5 is the proband’s sister, who had amenorrhea and infertility. Data for Subject IV-4, another sister of the proband, are from tests performed on day 25 of her menstrual cycle. The pedigree is shown in Figure 2. * The mean (±SD) value for castrates in the T assay is 0.25 ± 0.03 ng/mL. From [46] with permission.

**Table 2 ijms-22-12735-t002:** Natural FSHR mutations.

Patient/FSHR Mutation *	Puberty	Amenorrhea	Ovary	Ovarian Histology	Receptor Function In Vitro
Paris 1 ^a^/Pro519Thr	Delayed	Primary	Hypoplasic	↑Primordial and primary follicles	Total loss of function
Finnish 1 ^b^/Ala189Val	Delayed	Primary	Hypoplasic	↓Primordial; ↓secondary rare mature follicles	c-AMP: ∼29% of WT FSHR; IP3: almost totally lost
Paris 2 ^c^/Asp224Val, Leu601Val	Normal	Primary	Normal	Primordial, primary, secondary and antral follicles up to 3 mm diameter	c-AMP: ∼4% and 12% of WT FSHR
Paris 3 ^d^/Ile160Thr, Arg573Cys	Normal	Secondary	Normal	Primordial, primary, secondary and antral follicles up to 5 mm diameter	c-AMP: ∼9% and 24% of WT FSHR
Animal model					
FSHR KO mice ^e^			Hypoplasic	Primordial, primary, secondary follicles	Total inactivation

WT FSHR, wild-type FSH receptor. * References: ^a^ Meduri et al. [36]; ^b^ Aittomäki et al. [29]; Rannikko et al. [104]; ^c^ Touraine et al. [112]; ^d^ Beau et al. [111]; ^e^ Dierich et al. [96]. ↑: increased number; ↓: decreased number.

## Data Availability

See original references of the studies.

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
