# Peer review of "The Roles of Luteinizing Hormone, Follicle-Stimulating Hormone and Testosterone in Spermatogenesis and Folliculogenesis Revisited"

_ijms, 2021, doi:10.3390/ijms222312735_

Round 1

Reviewer 1 Report

The work is interesting but should be focused only on the role of hormones (FSH, LH, T) in spermatogenesis. The introduction in fact is focused only on the male side. The data are consistent.  The activity of hormones in the ovogenes is redundant and out of the scope of this review. This may be the subject of an other paper.

Author Response

Referee 1

The work is interesting but should be focused only on the role of hormones (FSH, LH, T) in spermatogenesis. The introduction in fact is focused only on the male side. The data are consistent.  The activity of hormones in the ovogenes is redundant and out of the scope of this review. This may be the subject of an other paper.

RESPONSE:

We have completed the introduction of the review as suggested by the Reviewer. We would like to highlight the fact that, up to now in all reviews or articles, the role of FSH is constantly described from the antral stage onwards, because all the authors refer on rodent but not humans models. It is thus important to highlight the differences between human and animal models and care should be taken when a comparison if done. We think that the clarification of the role of FSH in humans, based on the observations of mutated patients, is necessary. The role of FSH on the small follicular growth at the primordial stage has major consequences in therapeutics for small follicular maturation. 

Reviewer 2 Report

This review focuses on the effects exerted by LH, FSH and Testosterone on the correct development of male (and female) gonads and the associated production of functional sperm.

In addition, authors described interesting cases of partial or total absence of LH on the development of sub-fertility/infertility of patients, shedding light to this up-to-now not complete characterization of the complex mechanism regulating gonads and germinal cells development.

This review is very interesting and well written, with only few observations which would improve its quality.

Major:

Figure 3: the anti-hCG-beta antibody would bind to the beta subunit of hCG and not to the specific beta subunit of LH. Authors should better explain why they used this antibody and the limits that this use would infer.

The sentence: “Panel 2C shows markedly lower levels of mutant LH bioactivity in HEK 293 cells expressing the human LH receptor.” lacks several steps, which should describe how authors would reach these conclusions. Methods and background descriptions of the experiments carried out in this figure are missing

Minor:

  • The endocannabinoid system (ECS) has been closely associated with the hypothalamic-pituitary-gonadal (HPG) axis pathway, since cannabinoids receptors have been shown to interplay with hPG at multiple levels. For instance, they are differently expressed in the anterior pituitary Leydig cells and Sertoli cells (CB1), or in Sertoli cells (CB2), and FSH can induce hydrolysis of cannabinoids derivatives leading to apoptosis. A short paragraph should be added to also describe this regulation of the two hormones.
  • Other minor comments are in the pdf file attached

Author Response

Referee 2

This review focuses on the effects exerted by LH, FSH and Testosterone on the correct development of male (and female) gonads and the associated production of functional sperm.

In addition, authors described interesting cases of partial or total absence of LH on the development of sub-fertility/infertility of patients, shedding light to this up-to-now not complete characterization of the complex mechanism regulating gonads and germinal cells development.

This review is very interesting and well written, with only few observations which would improve its quality.

Major:

Figure 3: the anti-hCG-beta antibody would bind to the beta subunit of hCG and not to the specific beta subunit of LH. Authors should better explain why they used this antibody and the limits that this use would infer.

RESPONSE: We thank the Reviewer for the need of a clarification. In fact the in vitro study of the mutated LH is done by cotransfection of an expression vector encoding the mutated LHβ subunit and the α subunit in eukaryotic cells. As there is a strong identity between LHβ and hCG β, those anti-hCG- β antibodies are able to recognize the β subunit of LH with high efficiency. Furthermore hCG β antibodies are more efficient than LH β antibodies.

We have completed the sentence ine the legend of Fig 3: “Wild-type LH beta and mutant LH beta were immunodetected with an anti–human chorionic gonadotropin β antibody (Abcam) displaying strong cross reactivity”

The sentence: “Panel 2C shows markedly lower levels of mutant LH bioactivity in HEK 293 cells expressing the human LH receptor.” lacks several steps, which should describe how authors would reach these conclusions. Methods and background descriptions of the experiments carried out in this figure are missing

RESPONSE: In fact this sentence is the title of the paragraph and the conclusion of the experiments  described after in panel C of the figure legend. To make it clearer we have written in bold the title of all panels:

Panel A shows low intracellular and secreted levels of mutant LH beta subunit in transfected cells.

Panel B shows low levels of dimerization of the alpha subunit and mutant beta subunit of LH.

Panel C shows markedly lower levels of mutant LH bioactivity in HEK 293 cells expressing the human LH receptor

 We hope it is now clear for the reader.

Minor:

  • The endocannabinoid system (ECS) has been closely associated with the hypothalamic-pituitary-gonadal (HPG) axis pathway, since cannabinoids receptors have been shown to interplay with hPG at multiple levels. For instance, they are differently expressed in the anterior pituitary Leydig cells and Sertoli cells (CB1), or in Sertoli cells (CB2), and FSH can induce hydrolysis of cannabinoids derivatives leading to apoptosis. A short paragraph should be added to also describe this regulation of the two hormones.

REPONSSE: We understand the point of the reviewer, but because out text is focused on gonadotropin action, we think expanding the text on cannabinoids would be out of focus.

  • Other minor comments are in the pdf file attached

RESPONSE: We have corrected those minor points as indicated by the reviewer.